

# RapidHRV: an open-source toolbox for extracting heart rate and heart rate variability

Peter A. Kirk[1,2], Alexander Davidson Bryan[3], Sarah N. Garfinkel[1] and Oliver J. Robinson[1,4]

[1] Institute of Cognitive Neuroscience, University College London, University of London, London, United Kingdom
[2] Experimental Psychology, University College London, University of London, London, United Kingdom
[3] Independent Scholar, London, United Kingdom
[4] Clinical, Educational and Health Psychology, University College London, University of London, London, United Kingdom

## ABSTRACT

Heart rate and heart rate variability have enabled insight into a myriad of psychophysiological phenomena. There is now an influx of research attempting using these metrics within both laboratory settings (typically derived through electrocardiography or pulse oximetry) and ecologically-rich contexts (*via* wearable photoplethysmography, *i.e.*, smartwatches). However, these signals can be prone to artifacts and a low signal to noise ratio, which traditionally are detected and removed through visual inspection. Here, we developed an open-source Python package, RapidHRV, dedicated to the preprocessing, analysis, and visualization of heart rate and heart rate variability. Each of these modules can be executed with one line of code and includes automated cleaning. In simulated data, RapidHRV demonstrated excellent recovery of heart rate across most levels of noise ($>=10$ dB) and moderate-to-excellent recovery of heart rate variability even at relatively low signal to noise ratios ($>=20$ dB) and sampling rates ($>=20$ Hz). Validation in real datasets shows good-to-excellent recovery of heart rate and heart rate variability in electrocardiography and finger photoplethysmography recordings. Validation in wrist photoplethysmography demonstrated RapidHRV estimations were sensitive to heart rate and its variability under low motion conditions, but estimates were less stable under higher movement settings.

## RAPIDHRV: AN OPEN-SOURCE TOOLBOX FOR EXTRACTING HEART RATE AND HEART RATE VARIABILITY

Evidence has outlined a link between heart rate, heart rate variability, and health-related risks, ranging from cardiac mortality to mental illness (*Hillebrand et al., 2013*; *Jandackova et al., 2016*; *Makovac et al., 2016a*; *Pham et al., 2021*). Consequently, there is now an influx of research looking into whether these measures can be derived in

Corresponding author
Peter A. Kirk, p.kirk@ucl.ac.uk

naturalistic settings to track clinically-relevant outcomes, namely through wearable devices (*Georgiou et al., 2018*; *Mulcahy et al., 2019*). However, a key issue when opting to use the measures in naturalistic settings are the low signal to noise ratios (*e.g.*, photoplethysmography (PPG), a typical measure for cardiac monitoring in wrist wearables, *Caizzone, Boukhayma & Enz, 2017*). Moreover, heart rate variability measures generally require relatively longer windows for extraction compared to heart rate (*Baek et al., 2015*). Thus, significant noise poses a problem for out-of-laboratory applications, as point estimates can contain large errors from technological limitations and motion artifacts within windows of extraction.

In experimental settings, noise has often been dealt with through visual inspection of data (*Makovac et al., 2016b*; *Rae et al., 2020*); but when approaching time courses in relatively larger-scale samples, manual outlier detection is not a pragmatic solution.

Whilst some open-source packages are already available for the analysis of heart rate and heart rate variability, these are typically modality-specific, and not targeted at wrist-worn measures (*e.g., pyVHR* for video-based estimation, *Boccignone et al., 2020*). Some modality-general packages do exist, but these often still require manual visual inspection and/or can require custom scripting on the users end for tailoring to *e.g.*, noisy, wrist-worn PPG measures ('Analysing_Smartwatch_Data' in *HeartPy*, *van Gent et al., 2019*; *NeuroKit2*, *Pham et al., 2021*). As such, these are often less suited for dealing with datasets collected across large time frames. Consequently, we set out to develop a simple yet flexible toolbox for the extraction of time-domain heart rate and heart rate variability measures with automated artifact rejection applicable across recording modalities, including wrist-worn PPG. Here, we present the development and validation of an open-source Python package, 'RapidHRV'.

## PIPELINE

RapidHRV was developed in Python (V 3.9). RapidHRV source code and tutorials are available to download through PyPi (https://pypi.org/project/rapidhrv/) and GitHub (https://github.com/peterakirk/RapidHRV). Below we provide an overview of RapidHRVs preprocessing, analysis (Fig. 1), and visualization. Each of the three modules only requires one function (one line of code) to run, for which we have embedded examples at the end of the relevant sections below.

### Preprocessing

First, data is upsampled with cubic spline interpolation (3rd order polynomial; default = 1 kHz) to increase temporal accuracy of peak detection. To mitigate potential long-term drifts in the signal, the pipeline then applies a high pass butterworth filter (0.5 Hz) across the input data. Finally smoothing with a savitzky-golay filter (3rd order polynomial; default = 100 ms) is applied to reduce spiking (sharp increases in the signal caused by artifacts such as motion) whilst retaining temporal precision.

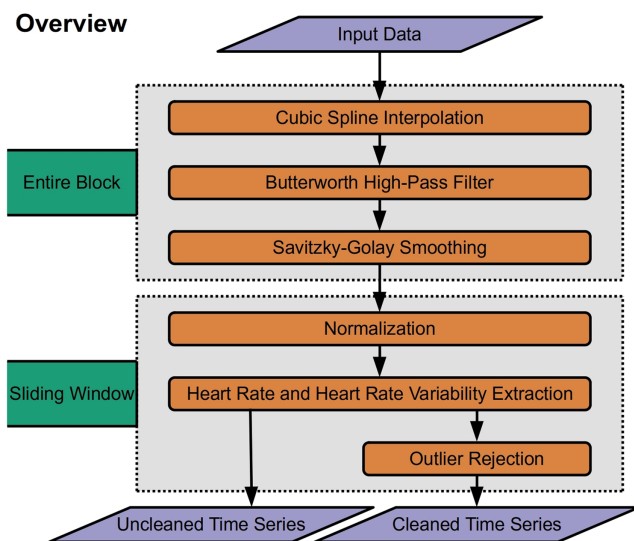

**Figure 1 Overview of RapidHRV pipeline.** Across an entire block, the pipeline initially processes data with high-pass filtering, upsampling, and smoothing. RapidHRV then applies a sliding window across the entire block. Within each window, the data are scaled. Heart rate (beats per minute) and heart rate variability (root mean squared of successive differences + standard deviation of intervals) are calculated for each window, and data is submitted to outlier rejection. RapidHRV produces both a cleaned and uncleaned time series of heart rate and heart rate variability.

# EXTRACTING HEART RATE AND HEART RATE VARIABILITY

Following preprocessing, the pipeline scales the data (between 0 and 100) and runs peak detection on every window (default width = 10 s; for a methodological discussion and prior validation of using ultra-short, 10 s windows in heart rate variability estimation, see *Munoz et al., 2015*). This outputs peaks and their properties (*e.g.*, heights, amplitudes, width; *SciPy* 'find_peaks', *Virtanen et al., 2020*). For ECG data however, peak detection is vulnerable to irrelevant prominent P and T waves. Specifically, traditional amplitude-based analyses may occasionally detect non-R wave peaks that demonstrate a similar or greater amplitude than R waves. Consequently, for ECG data, RapidHRV can implement k-means clustering (k = 3) to discern R waves from P and T waves prevalent in the signal (*scikit-learn* 'KMeans', *Pedregosa et al., 2011*). This is implemented by reducing the minimum amplitude threshold to near-zero (*i.e.*, 5%), running amplitude-based peak detection, then sorting peaks into three clusters using relevant properties (*i.e.*, peak widths, heights, and prominences). R waves are then determined based on cluster centroids for peak properties, expecting R waves to hold higher prominences and lower widths compared to P and T waves. Figure 2 demonstrates an example (from dataset 3, see *Validation Methods*) wherein amplitude-based analyses may incorrectly identify T waves as R waves in an atypical ECG signal, and how RapidHRV's peak clustering helps mitigate this.

As RapidHRV uses fixed movements for the sliding window, a window can start or end at any point during the cardiac signal. This can occasionally result in underestimation of
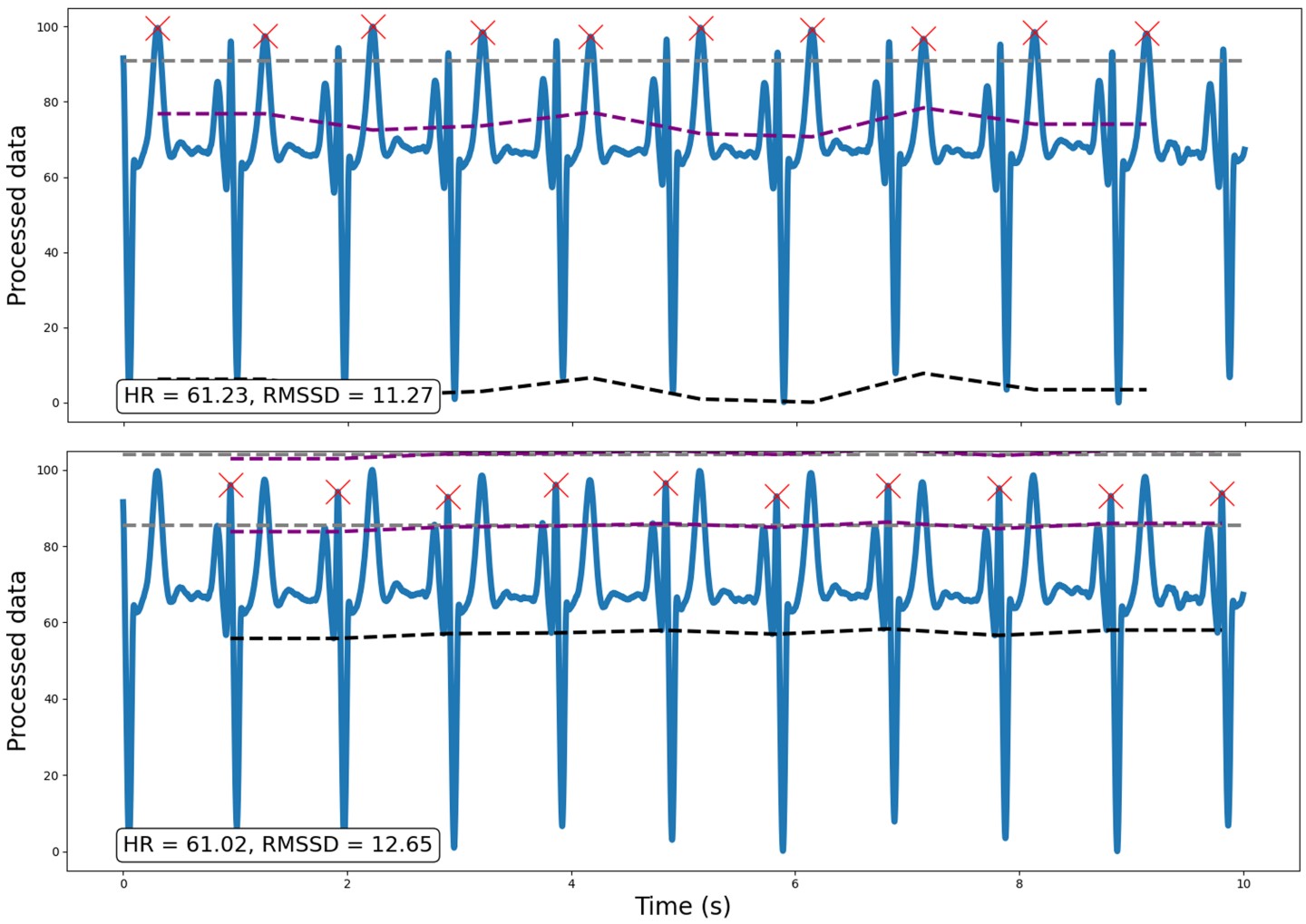

**Figure 2 Example of how the implementation of traditional amplitude-based analyses may be insufficient for peak detection, particularly for atypical ECG signals.** Traditional amplitude-based peak detection (top) incorrectly identifies T waves as the relevant peak of interest. However, using RapidHRV's implementation of k-means clustering (bottom), we are able to derive measures based on the correct peaks (*i.e.*, R waves).

the first/last peak's amplitude as the baseline value may—for example—be set during the P wave. Therefore, RapidHRV recalculates amplitudes of the first/last peaks using baseline imputation from the neighboring peaks.

For extracting beats per minute (BPM) the number of peaks, $k - 1$, is multiplied by 60 (seconds), and divided by the difference in time between the first and last peaks, i and j:

$$BPM = ((k - 1) * 60)/(j - i)$$

The root mean square of successive differences is calculated by obtaining: (1) the interbeat interval, $IBI_i$, between neighboring peaks; (2) the successive differences in interbeat intervals, $IBI_i - IBI_{i+1}$; (3) the square of differences; (4) the mean of squared differences (dividing by the number peaks, $N - 1$); and (5) the root of the mean square of successive differences (RMSSD):

$$\text{RMSSD} = \sqrt[2]{\frac{\sum_{i=1}^{N-1}(\text{IBI}_i - \text{IBI}_{i+1})^2}{N-1}}$$

BPM and RMSSD were selected as the primary measures as they appear to be the most stable metrics when derived from ulta-short recordings (*Baek et al., 2015*). RapidHRV also supplements these measures with the standard deviation of N-N intervals (SDNN), standard deviation of successive differences (SDSD), proportion of successive differences greater than 20 ms (PNN20), proportion of successive differences greater than 50 ms (PNN50), and high-frequency power (HF; note: as this requires more data points than time-domain analyses NaN is returned if there is insufficient data).

## Outlier detection

The last phase of the pipeline is to pass measures derived from peak extraction to outlier rejection (Fig. 3). This is applied at the level of the sliding window. If a window is declared an outlier, heart rate and heart rate variability measures are removed from the cleaned time series. By default, RapidHRV returns both the cleaned and the uncleaned time series. In addition to default parameters listed below, the package has optional arguments embedded to allow users to override these presets. Given that not all users may be entirely comfortable manually adjusting these, RapidHRV additionally contains semantically-labeled arguments as inputs for outlier constraints ('liberal', 'moderate' [default], and 'conservative'; corresponding parameters are parenthesized under *Biological Constraints* and *Statistical Constraints*).

**Biological Constraints**. RapidHRV first applies restrictions to exclude data that are highly unlikely given known physiology:

1. Screening for sufficient peaks in a window (default: number of peaks > (window width/5) + 2), floored at 3; default at 10 s = 3 peaks). This is primarily for computational applicability and efficacy, screening data prior to further processing. The minimum number of peaks required to enable calculation of RMSSD is 3. As such, this is also applied to the uncleaned time series.

2. Minimal and maximal heart rate ('moderate' [default]: 30 > BPM > 190; 'liberal': 20 > BPM > 200; 'conservative': 40 BPM > 180). These boundaries were based on typical heart rate at rest and during exercise in the healthy population (*Pierpont & Voth, 2004*; *Sandvik et al., 1995*; *Savonen et al., 2006*).

3. Minimal and maximal heart rate variability ('moderate' [default]: 5 > RMSSD > 262; 'liberal': 0 > RMSSD > 300; 'conservative': 10 > RMSSD > 200). Default arguments correspond to the minimum/maximum 2nd/98th percentiles of resting RMSSD across ages 16–89 years (*van den Berg et al., 2018*).

**Statistical constraints**. RapidHRV next applies statistical constraints to account for noisy data that may otherwise appear to provide measures within the range of known physiology:
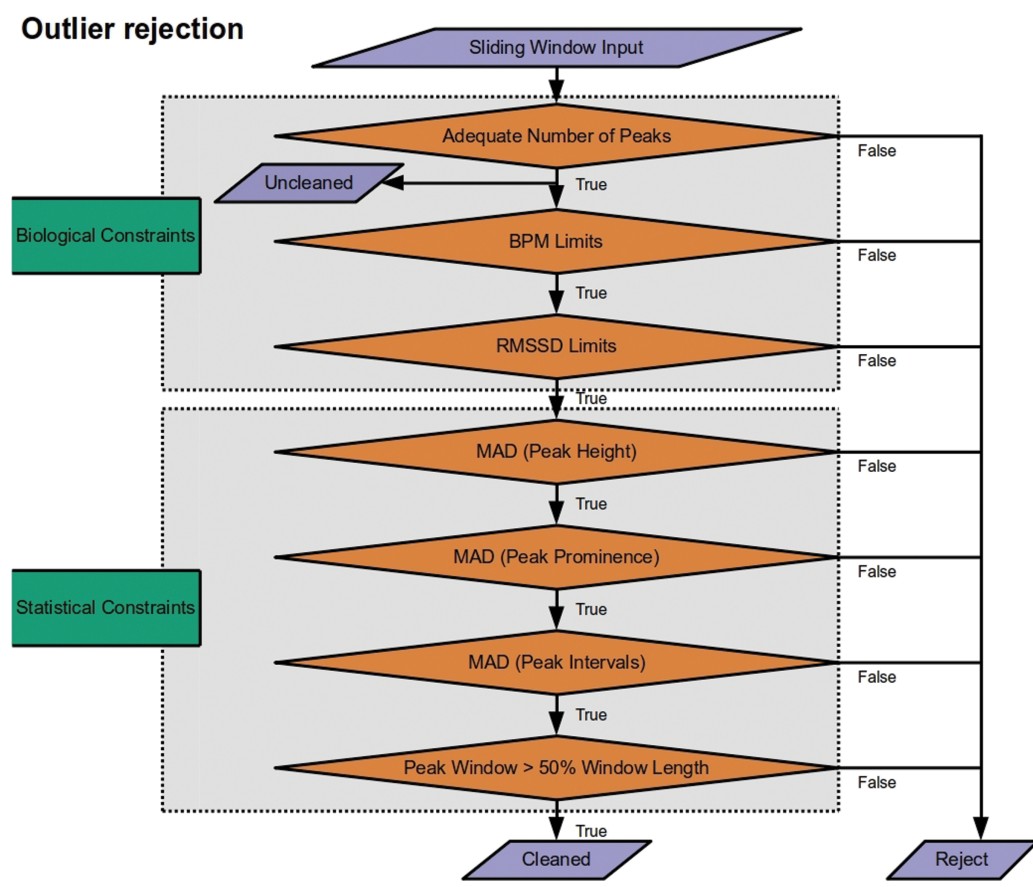

**Figure 3 Outlier rejection methods for Rapid HRV.** The only outlier rejection method applied to the uncleaned time series is screening for a sufficient number of peaks to derive metrics. The cleaned time series then goes through a battery of biological constraints (thresholding minimum/maximum beats per minute (BPM) and root mean square of successive differences (RMSSD)) and statistical constraints (median absolute deviation (MAD) of peak heights, prominences, and intervals; ensuring adequate duration from first to last peaks).

4. Median absolute deviation (MAD) of peak heights (distance from minimum value of signal in window; *i.e.*, 0) and prominence (amplitude from baseline height; 'moderate' [default] = 5 MAD units; 'liberal' = 7; 'conservative' = 4). Unlike Z-scoring, this quantifies each peak's height and prominence in a given window in terms of its deviation from the median value in the same window (for a discussion of median absolute deviation see *Leys et al., 2013*). Applying these constraints to height and prominence helps exclude windows with noise-driven inaccuracies in peak detection.

5. Median absolute deviation of interbeat intervals ('moderate' [default] = 5 MAD units; 'liberal' = 7; 'conservative' = 4). This was also implemented to account for inaccuracies in peak detection, either where spiking may cause detection of an irrelevant peak shortly after *e.g.*, an R wave, or low signal to noise ratio may result in missing relevant peaks.

6. Time from the first peak to the last peak does not recede 50% of the fixed window width. This is to ensure the user that the *actual* length of time for extracting HR/HRV is

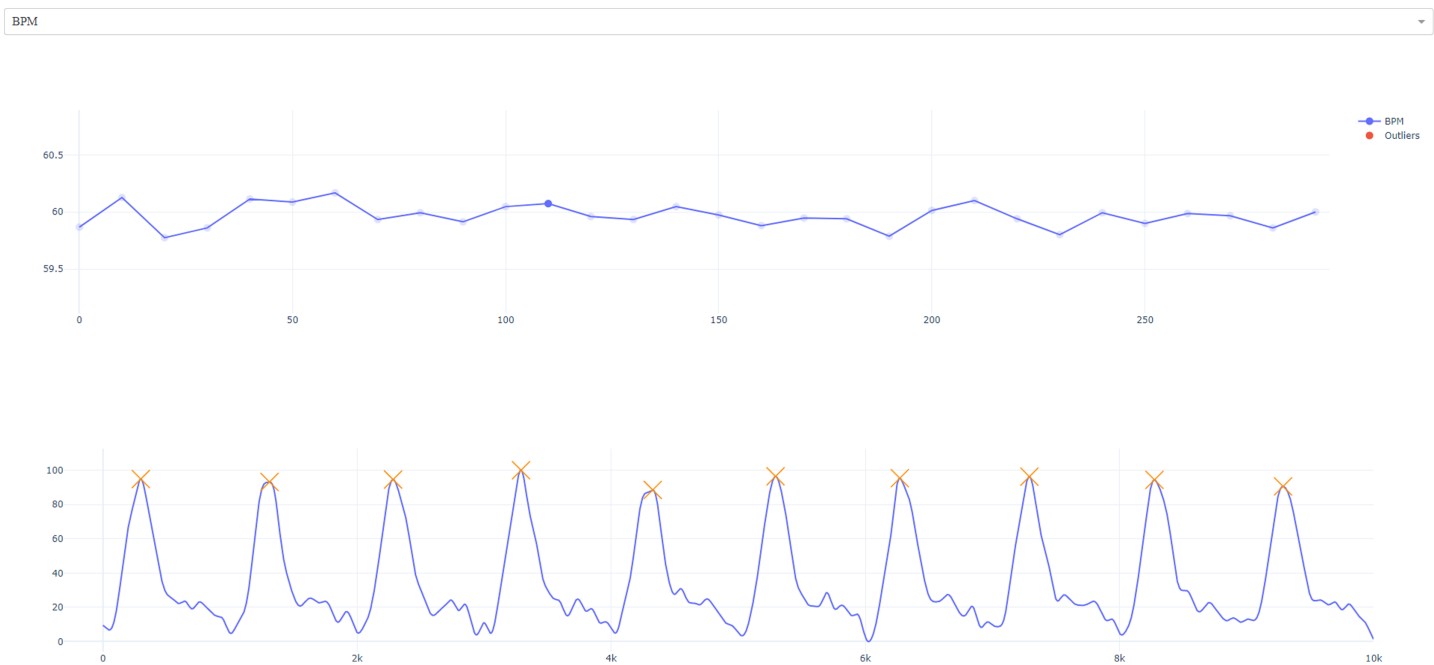

**Figure 4 Interactive visualization.** The user is presented with the timeseries of HR/HRV metric (specific measure can be selected from the top tab; *i.e.*, RMSSD has been selected). Users can click on specific timepoints to view extraction windows (top right; foreground).

not less than half of that which is specified in the window width argument. Given debates surrounding adequacy of different window lengths for HRV extraction (*Baek et al., 2015*; *Munoz et al., 2015*), this was implemented primarily as a theoretical constraint (rather than for just cleaning *per se*) to ensure the user is not provided data that deviated significantly from their specified window.

Analysis can be executed with one line of code, which returns a pandas DataFrame (*McKinney, 2012*; *Reback et al., 2022*) containing the analyzed data.

## Visualization

To allow for selected manual inspection, we have also implemented optional interactive visualizations *via* matplotlib (*Hunter, 2007*) which allow the user to plot the time course of heart rate and heart rate variability. The user can then select and view specific data points to see the window of extraction (Fig. 4). We have provided an example pipeline below:

```
import rapidhrv as rhv
example_signal = rhv.get_example_data() # Load example data
preprocessed = rhv.preprocess(example_signal) # Preprocess data
analyzed = rhv.analyze(preprocessed) # Analyze data
rhv.visualize(analyzed) # Visualize data
```

**Table 1 Simulated and real datasets used for validation of RapidHRV.**

| Dataset | Modality | N | Conditions | Duration | Hz |
|---------|----------|---|------------|----------|-----|
| 1. Simulation | PPGSynth | 273 | None (noise) | 5 min | 20–250 Hz |
| 2. Simulation | PPGSynth | 172 (10 repeats) | 'Anxiety' | 5 min | 20–250 Hz |
| 3. *Iyengar et al. (1996)* | ECG | 40 | Movie | ~2 h | 250 Hz |
| 4. *de Groot, Kirk & Gottfried (2020)* | Finger IR PPG | 39 | Anxiety (sitting) | 2 × 28 min | 1,000 Hz |
| 5. *Reiss et al. (2019)* | Wrist PPG | 15 | Varying activities | ~2.5 h | 64 Hz |

# VALIDATION METHODS

## Datasets

To validate the above pipeline we subjected it to a series of tests across both simulated and real data (Table 1). We first started by testing RapidHRV's estimations in two sets of simulated data (PPGSynth; *Tang et al., 2020*). Next, we ran validation in real data across successively noisier modalities: electrocardiography (ECG), finger infrared photoplethysmography (IR PPG), and wrist PPG data. Database information and code generated in validation tests are available through the open science framework (https://osf.io/7zvn9/).

**1. Simulations Across Noise and Sampling Rates**. We first took the pipeline forward to validation using simulated photoplethysmography (PPG) data from PPGSynth (*Tang et al., 2020*) in MATLAB. This allowed us to test how well RapidHRV recovered known parameters under specified conditions, such as sampling rate and noise. We produced 5 min long synthetic datasets (1,000 Hz), each of which varied according to the following cardiac features:

- Mean heart rate (BPM range 60–120, increments of 5). These were selected to allow us to check the sensitivity of the pipeline for detecting values across typical resting heart rate, as well as elevations of these values.
- Heart rate variability (RMSSD range 0–100, increments of 5). Again, these were selected based on typical resting heart rate variability and moderate increases/decreases.

 Following simulation of the data, we introduced noise *via*:

- White gaussian noise filtering (signal to noise ratios = 0.01, 10, 20, 30, 40, 50, and 60 dB). Here, we selected a range of values starting from near-zero to ascertain at which level of noise the pipeline could no longer recover parameters (with near-zero expected to prevent the extraction of any meaningful metrics).
- Downsampling (frequencies = 20, 50, 100, and 250 Hz). These were selected: (A) to capture the range of sampling rates currently used in photoplethysmography studies (typically >= 20 Hz); and (B) because prior work has suggested sensitivity to RMSSD deteriorates dramatically between 20–100 Hz (*Choi & Shin, 2017*). This offered the opportunity to test the effects of cleaning on what would otherwise be considered poor quality data.

**2. Simulated 'Anxiety'.** Again, we simulated data using PPGSynth. However, simulations were based on prior anxiety literature to emulate experiment-specific effects. We wanted to validate the sensitivity of the pipeline across specified sampling rates and levels of noise but in the context of a known psychophysiological effect. For this, we used an estimated Cohen's *d* of 0.384, which was derived from a previous within-subjects threat-of-shock study (RMSSD, $t_{25} = 1.96$, polarity flipped for readability; *Gold, Morey & McCarthy, 2015*). To reflect decisions in experimental design, we ran a power calculation using the 'pwr' package (*Champely et al., 2017*) in R (*R Core Team, 2013*), leading us to simulate a sample of $N = 171$ ($\alpha = 0.001$, $1 - \beta = 0.95$). These 'experiments' were simulated 10 times to ascertain confidence intervals around estimated effect sizes.

For each 'subject', we generated a 5 min simulated time series (1 kHz) using typical resting heart rate and heart rate variability (BPM: $\mu = 74$, $\sigma = 13$, *Savonen et al., 2006*; RMSSD: $\mu = 23$, $\sigma = 7$, bounded between 5–262, *Nunan, Sandercock & Brodie, 2010*). These simulations were intended to emulate the 'safe' condition (no anxiety induction) in a threat-of-shock study. Next, we simulated another time series for each subject that deviated from their 'safe' RMSSD (heart rate held constant) based on our effect size estimate ($d = 0.384$). This emulated our 'threat' (anxiety induction) condition and contained general reductions in heart rate variability. Each time series was finally submitted to downsampling and noise filtering using the same parameters as in the previous simulations (white gaussian noise filtering to 0.01, 10, 20, 30, 40, 50, 60 dB; downsampling to 20, 50, 100, and 250 Hz).

**3. Estimation *via* ECG.** Following simulations, we wanted to clarify that our package was able to adequately extract measures from one of the highest standards for heart rate variability recordings, ECG. Here, we tested whether our package was able to recapitulate known age-related effects of heart rate variability. For this, we used the *Fantasia* database (*Iyengar et al., 1996*). This consisted of 40 subjects (20 'Young' Age Range = 21–34 years; 20 'Old' Age Range = 68–81 years) watching the movie 'Fantasia' (duration ≈ 2 h) whilst undergoing ECG recordings. Automated peak detection was previously run on this dataset, with every beat annotation verified by visual inspection. The full dataset and description is available on PhysioNet (https://physionet.org/content/fantasia/1.0.0/; *Goldberger et al., 2000*).

**4. Estimation *via* finger PPG.** We next took the pipeline forward to validation in a modality considered relatively noisier than ECG, finger photoplethysmography. Here, we used a dataset of 39 subjects (Age: Mean = 22.67; Range = 18–38 years; demographics reported prior to $N = 1$ exclusion) watching $2 \times 28$ min blocks of documentary and horror video clips undergoing finger IR PPG recording (1,000 Hz, *de Groot, Kirk & Gottfried, 2020*). This allowed us to contrast psychological conditions of the experiment, testing whether RapidHRV was able to detect effects of anxiety. Moreover, in the original study, data had been preprocessed and analyzed using a commercially available software (Labchart; ADInstruments, Sydney, Australia; analyzed using built-in 'Peak Analysis' module). This allowed us to benchmark RapidHRV against another software. The full dataset and description is available *via* the Open Science Framework (https://osf.io/y76p2/).

**5. Estimation *via* wrist PPG**. Finally, we analyzed the *PPG-DaLiA* dataset, which consists of 15 subjects (Age: Mean = 30.60 years; Range = 21–55) completing various activities whilst having wrist PPG recorded with an Empatica E4 device (Hz = 64) and simultaneous ECG measures with a RespiBAN (Hz = 700, *Reiss et al., 2019*). Over the course of 2.5 h, participants engaged in a range of activities designed to elicit low and high motion. These were: sitting and reading; ascending/descending stairs; 1 v 1 table soccer; cycling on pavements and gravel; driving a car; lunch break (queuing, purchasing, and eating food in a cafeteria); walking; and working at a desk (typically on a computer). This allowed us to test whether RapidHRV was able to extract heart rate and variability measures from wrist PPG and how these compared to ECG measurements. Moreover, this enabled us to highlight under what conditions estimations are optimal. For the full dataset and description, see *Reiss et al. (2019)*.

## Analyses

Unless otherwise stated, all analyses were conducted using RapidHRV's default arguments: window width = 10 s; window movement = 10 s; outlier method = 'moderate' ((peak/height median absolute deviation = 5, interbeat interval median absolute deviation = 5, BPM range = 30–190, RMSSD range = 5–262); minimum window successful extraction = 5 s, minimum amplitude for peak detection = 50, minimum distance between peaks = 250 ms; for ECG data, ecg_prt_clustering = True). To assess performance across datasets, we used: visualizations; intraclass correlation coefficients (ICC; two-way mixed effects, absolute agreement, single measure); root-mean-square-error (RMSE); and sensitivity to experimental effects (Cohen's *d*). For ICC values, we used the following semantic labels for interpretation: ICC < 0.5 as 'poor', 0.5 < ICC < 0.75 as 'moderate'; 0.75 < ICC < 0.90 as 'good', and 0.90 < ICC as 'excellent' (*Koo & Li, 2016*).

In our wrist PPG dataset, we also calculated motion estimates as a proxy for the severity of noise present in PPG across conditions. We derived mean jerk magnitude (square root of the sum of squared changes in acceleration; *Eager, Pendrill & Reistad, 2016*), which aimed to pick up on oscillatory ('jerk') motions in the wrist for each condition. Across all axes (x, y, and z; m/s^2), jerk vector, j, was calculated as the change in acceleration, a, divided by the change in time, t, between samples:

$$\vec{j}(t) = \frac{\Delta\vec{a}(t)}{\Delta t}$$

Finally, we derived jerk magnitudes by calculating the square root of the sum of squared jerks, j, for axes x, y, and z:

$$|\vec{j}| = \sqrt{j_x^2 + j_y^2 + j_z^2}$$

For each subject, jerk magnitudes were averaged across time for each condition.

## VALIDATION RESULTS

### Parameter recovery in simulated PPG

RapidHRV was able to accurately recover heart rate across most sampling frequencies and noise in our initial simulations. Accurate detection of BPM primarily started to degrade when signal to noise ratios were less than 10 dB (Fig. 5; Table 2). RapidHRV cleaning provided improvements in simulations with a signal to noise ratio of 10 dB.

Performance in recovery of heart rate variability was again primarily based on signal to noise ratio. At 20 dB RapidHRV recovery of RMSSD was good-to-excellent for higher sampling rates (>=100 Hz), whereas lower sampling rates (<100 Hz) required slightly lower levels of noise (>30 dB) for excellent recovery. RapidHRV cleaning provided clear improvements when signal to noise ratio was below 30 dB (Fig. 3).

### *Key points*

- RapidHRV's estimation of simulated BPM was good-to-excellent when signal to noise ratio was 10 dB or above.
- RapidHRV's estimation of simulated RMSSD was good-to-excellent when signal to noise ratio was 20 dB and sampling rate was 100 Hz or higher. Across all sampling rates, RMSSD recovery was excellent at 30 dB or above.
- Cleaning appeared particularly helpful when noise was high and/or sampling rate was low.

### Sensitivity to 'Anxiety' in simulated PPG

RapidHRV's ability to estimate simulated effects was again primarily impacted by the level of SNR, where changes in RMSSD were not reliably detected at 0.01 and 10 dB (Fig. 6; Table 3). Effects were detected at signal to noise ratios of 20 and 30 dB, but this was estimated at around half that of the true value. Maximal effect sizes plateaued following 40 dB for uncleaned data and 30 dB for cleaned data. Moreover, reliable detection of effects in some scenarios (*e.g.*, 50 Hz, 10 dB) appeared dependent on cleaning. This was consistent with our previous validation results, in that cleaning was beneficial at lower signal to noise ratios and sampling rates, but was not necessary (or could be relaxed) in cleaner data.

### *Key points*

- RapidHRV's estimation for simulated effects of Anxiety on RMSSD was excellent when signal to noise ratio was 30 dB or above. Moreover, this estimation was robust at a low sampling rate (*i.e.*, 20 Hz).
- Cleaning was beneficial when noise was high, but was not necessary when noise was low.

### Estimation *via* ECG

As expected, visual inspection suggested ECG data to hold a high signal to noise ratio. In line with our simulation findings, we adjusted the outlier rejection method accordingly

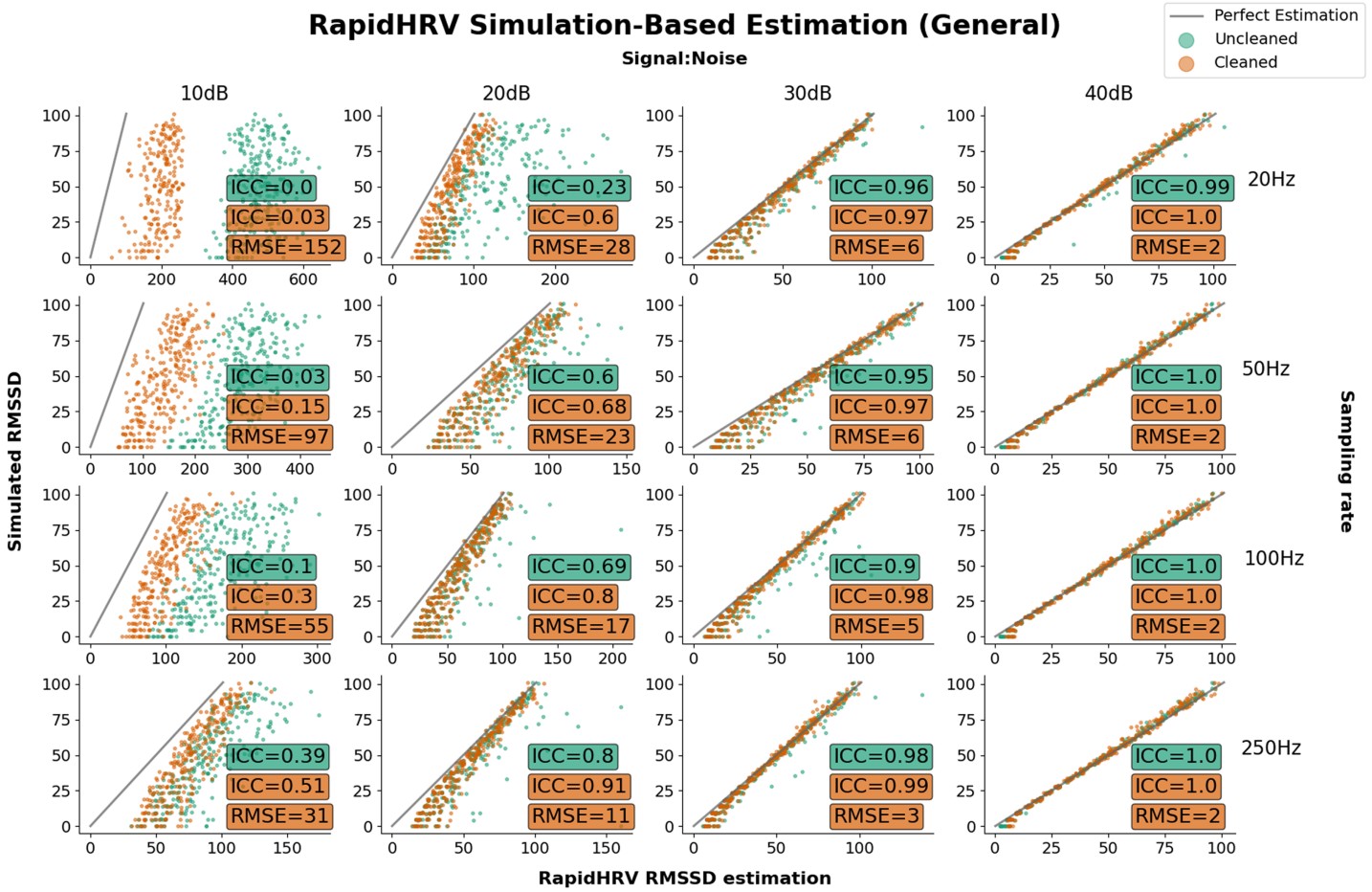

**Figure 5 Parameter recovery of simulated PPG data as a function of heart rate variability.** Parameter recovery of simulated PPG data as a function of heart rate variability (Intraclass Correlations (ICC) and Root Mean Square Error (RMSE) against ground truth). Y axes reflect the true RMSSD in the data, whilst the X axes reflect RapidHRV's estimation. For readability, data is only plotted in a key range of performance (subplots: 10–40 dB noise left-to-right; 20–250 Hz top-to-bottom).

so as not to excessively exclude data (*i.e.*, 'liberal'). The dataset had previously been analyzed using an automated peak detection algorithm, with every beat annotation verified by visual inspection (*Iyengar et al., 1996*). Subject-specific heart rate estimates were not available from the original database. However, RapidHRV heart rate estimations were able to recapitulate sample-wide summary statistics (Fig. 7; Table 4).

The analytical method used for extraction of heart rate variability in the original database (power spectral density analysis) was inconsistent with RapidHRV's time-domain heart rate variability measure (RMSSD). Despite this discrepancy, RapidHRV was able to capture previously reported effects of age on heart rate variability (*Iyengar et al., 1996*; estimated Cohen's *d* of short-term heart rate variability (*i.e.*, $\alpha_s$) ≈ 1.32), such that—in cleaned data—younger participants demonstrated higher RMSSD (*M* = 58.94, *SD* = 28.82) than older participants (*M* = 25.54, *SD* = 13.02, Cohen's *d* = 1.49; Fig. 6). Effects were not apparent in the uncleaned data (Cohen's *d* = −0.31).

**Table 2 Intraclass correlations between simulated BPM and RapidHRV for uncleaned (and cleaned) data as a function of sampling rate and noise.**

| Sampling rate | 0.01 dB | 10 dB | 20 dB |
|---|---|---|---|
| 20 Hz | 0.06 (*18*) | 0.37 (*0.80*) | 0.98 (*1.0*) |
| 50 Hz | 0.18 (*0.13*) | 0.80 (*0.96*) | 1.0 (*1.0*) |
| 100 Hz | 0.33 (*0.26*) | 0.96 (*1.0*) | 1.0 (*1.0*) |
| 250 Hz | 0.62 (*0.7*) | 1.0 (*1.0*) | 1.0 (*1.0*) |

**Note:**
Higher values indicate better performance. For readability, data is only reported in a key range of performance (0.01–20 dB noise) where ICC had plateaued at ~0 and ~1.

**Figure 6 Average estimated effect size of simulated 'anxiety'.** Average estimated effect size (+95% Confidence Intervals) of simulated 'anxiety' on the root mean square of successive differences as a function of signal to noise ratio (SNR), sampling rate, and cleaning. Higher values indicate greater sensitivity. This suggested RapidHRV was able to estimate true effect sizes in data when signal to noise ratio was >=30 dB. Visualized with ggplo2 (*Wickham, 2016*).

**Table 3 RMSE between ground truth and RapidHRV RMSSD collapsed across conditions.**

| | 0.01 dB | 10 dB | 20 dB | 30 dB | 40 dB | 50 dB | 60 dB |
|---|---|---|---|---|---|---|---|
| 20 Hz | 194 | 155 | 37 | 9 | 2 | 2 | 2 |
| 50 Hz | 189 | 110 | 29 | 8 | 2 | 2 | 2 |
| 100 Hz | 187 | 21 | 21 | 6 | 2 | 2 | 2 |
| 250 Hz | 171 | 14 | 14 | 3 | 2 | 2 | 2 |

## *Key points*

- In a movie-watching ECG dataset, RapidHRV was able to recapitulate previously reported summary statistics of BPM.
- RapidHRV was able to reproduce previously reported effects of age on RMSSD as measured by ECG.
- Cleaning was vital for detecting effects of age on RMSSD in ECG.

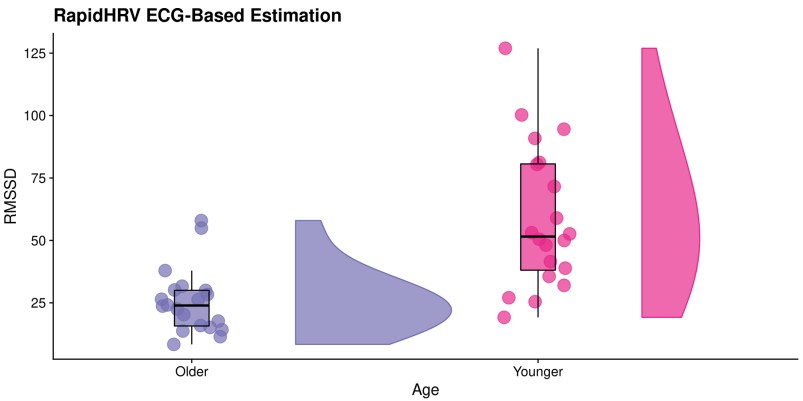

**Figure 7 Effects of age on RapidHRV-estimated RMSSD.** RainCloud plot (*Allen et al., 2019*) demonstrating effects of age on RapidHRV-estimated RMSSD (cleaned) derived from movie-watching ECG (*N* = 40). As expected, older participants typically had lower heart rate variability than younger participants.

**Table 4 Mean heart rate (and standard deviation) by age group and analysis.**

|  | *Iyengar et al. (1996)* | **RapidHRV (Cleaned)** |
| --- | --- | --- |
| Young | 60.55 (8.77) | 63.54 (9.51) |
| Old | 57.22 (8.60) | 58.50 (9.17) |

**Table 5 Effect size (Cohen's *d*) of heart rate and variability between conditions as a function of software and cleaning method.**

| Software | BPM effect (*d*) | RMSSD effect (*d*) |
| --- | --- | --- |
| Labchart | 0.52 | 0.19 |
| RapidHRV (Uncleaned) | 0.45 | −0.04 |
| RapidHRV (Cleaned) | 0.54 | 0.35 |

## Estimation *via* finger PPG

### Sensitivity to anxiety

In our finger IR PPG data, RapidHRV was able to capture previously reported (*de Groot, Kirk & Gottfried, 2020*) effects of anxiety on BPM (Table 5). RapidHRV additionally demonstrated an influence of anxiety on RMSSD. Effects on BPM were greater following cleaning, whereas detection of effects on RMSSD was entirely dependent on cleaning.

### Benchmarking

Overall, there was excellent agreement between RapidHRV and previous estimates (*de Groot, Kirk & Gottfried, 2020*; implemented using LabChart, ADInstruments, Sydney, Australia) of BPM (ICC > 0.99; Fig. 8). For heart rate variability, there was good agreement between the two when using the cleaned time series (ICC = 0.89), but poor agreement when using the uncleaned time series (ICC = 0.32).

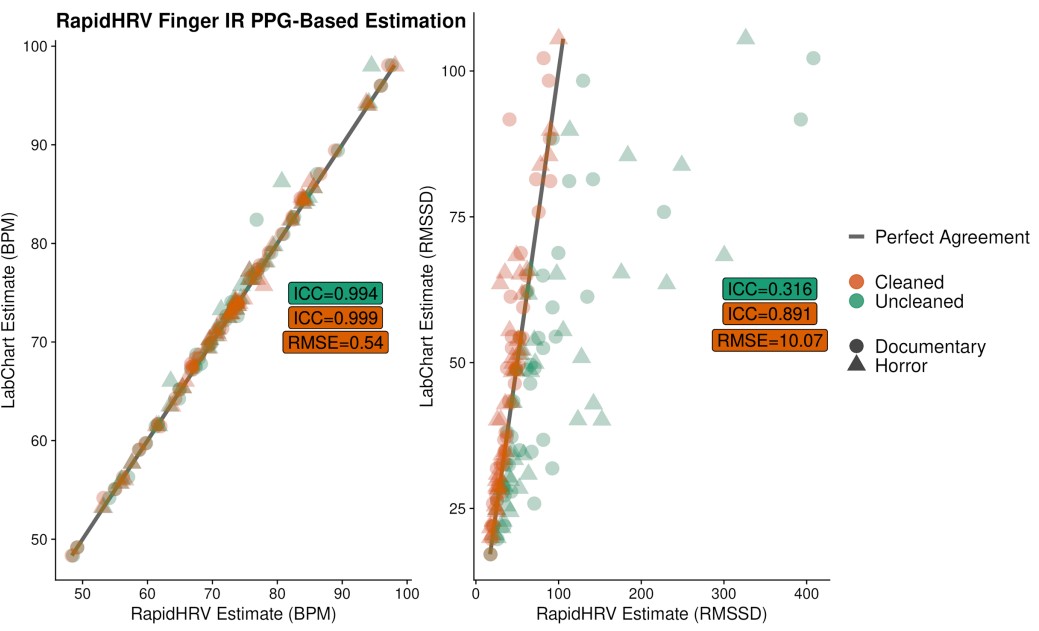

**Figure 8 Agreement between RapidHRV and a previous Labchart analysis.** Agreement between RapidHRV and a previous Labchart analysis (*de Groot, Kirk & Gottfried, 2020*) of heart rate and variability in a finger IR PPG dataset (*N* = 39).

### *Key points*

- RapidHRV was able to recapitulate previously reported effects of anxiety on BPM and RMSSD as measured by finger IR PPG.
- Cleaning was vital to the detection of effects of anxiety on RMSSD.

### Estimation *via* wrist PPG during activities

We first derived motion estimates (mean jerk magnitude; *Eager, Pendrill & Reistad, 2016*) as a proxy for the severity of noise present in PPG. This enabled us to split conditions into low (Reading, Working, Lunch Break, Driving) and high motion activities (Table Soccer, Stairs, Walking, Cycling; Fig. 9).

In our initial analysis using default arguments, RapidHRV was not able to produce estimates across many of the activities for both cleaned and uncleaned data (conditions estimated: $M = 22.5\%$, $SD = 5.18\%$). Visual inspection confirmed this was due to high levels of noise and variable peak amplitude present in the signal. Therefore, we adjusted window movement to 1 s (to increase the number of extraction windows in each condition), reduced minimum amplitude threshold for peak detection (30), and—in line with our simulation results above (Fig. 3)—tightened outlier rejection ('conservative'). Following this, RapidHRV was able to produce estimates for all conditions in the uncleaned data and across most activities for cleaned data (conditions estimated: $M = 87.5\%$, $SD = 11.57\%$). Missing estimates from the latter were predominantly limited to high motion activities (1 missing estimate for a single subject in the driving condition).

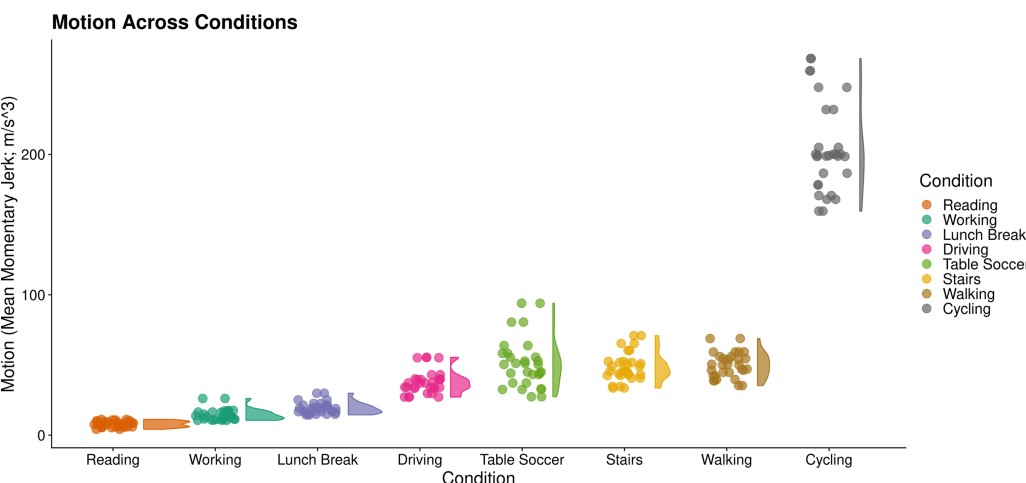

**Figure 9 Motion across wrist PPG activities.** RainCloud plot (*Allen et al., 2019*) of motion across activities. Each datapoint represents an individual subject's mean momentary jerk magnitude within a condition. Curves demonstrate distribution density across the sample (kernel density plot).

**Table 6 Intraclass correlations and RMSE between cleaned heart rate and variability measured derived from ECG R-R intervals (visually inspected) and RapidHRV PPG-based estimation.**

| Activity | BPM | | RMSSD | |
|---|---|---|---|---|
| Low motion | ICC | RMSE | ICC | RMSE |
| Reading | 1.00 | 0.57 | 0.61 | 30.02 |
| Driving | 0.99 | 2.81 | 0.68 | 24.68 |
| Lunch break | 0.94 | 4.45 | 0.58 | 31.88 |
| Working | 0.90 | 5.16 | 0.82 | 18.26 |
| High motion | | | | |
| Walking | −0.15 | 30.01 | −0.12 | 87.34 |
| Cycling | 0.75 | 17.83 | 0.27 | 46.93 |
| Table soccer | 0.04 | 25.29 | 0.32 | 45.97 |
| Stairs | 0.39 | 28.95 | −0.18 | 55.13 |

Across low motion activities, RapidHRV PPG-based estimates converged with simultaneous (visually verified) ECG measures (*Reiss et al., 2019*). Agreement with low motion activities was excellent for BPM (ICC > 0.90) and agreement with RMSSD was moderate-to-good (0.57 < ICC < 0.82; Table 6; Fig. 10). Under high motion conditions, heart rate and heart rate variability estimates showed poor agreement (ICC < 0.32), except for heart rate within the cycling condition (ICC = 0.75).

### *Key point*

- Following calibration to the data, cleaned RapidHRV wrist PPG-based estimates demonstrated moderate-to-excellent convergence with a simultaneous ECG-based analysis whilst under low motion conditions. Agreement was generally poor under high motion conditions.

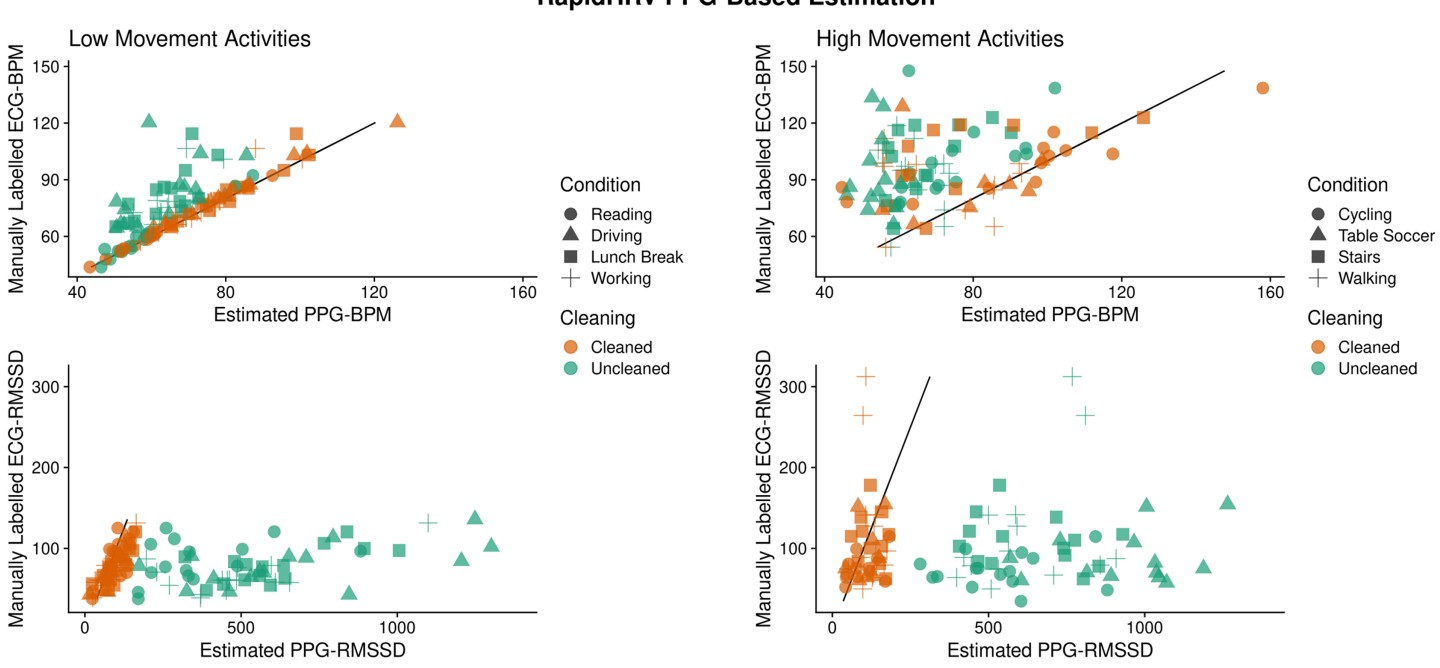

**Figure 10 Agreement between PPG-based RapidHRV-estimates and visually inspected ECG-based estimates across different activities.**

## DISCUSSION

RapidHRV is an open-source toolbox for extracting heart rate and heart rate variability measures. RapidHRV was developed in response to the need for software dedicated to dealing with extensive cardiac data collected across large time frames, such as out-of-laboratory PPG recordings, which may require point estimates from very short time windows (~10 s). Python packages currently exist which can analyze cardiac data (*e.g., Systole, Legrand & Allen, 2020; NeuroKit2, Pham et al., 2021; pyHRV, Gomes, Margaritoff & da Silva, 2019*). However, outlier rejection algorithms often require visual inspection and/or extensive scripting on the user's end. While suitable for the cardiac data collected during laboratory experiments, this may not be feasible when dealing with data collected across large time-scales, such as weeks or months. Here, we have attempted to fill this gap by developing a programmatically easy-to-use toolbox which extracts HRV measures from ultra-short windows and automates artifact detection and rejection. In general, this is applied *via* a series of biological and statistical constraints. Moreover, for ECG data, we have also implemented a k-means clustering algorithm for delineating P, R, and T waves. Across simulated and real datasets, we scrutinized RapidHRV, testing scenarios where it was and was not able to extract meaningful metrics. We show that signal to noise ratio, sampling rate, and recording modality had a clear impact on sensitivity of estimation. Here, we summarize these validation tests and make modality-specific recommendations for users.

## Simulations

Within simulated data, RapidHRV was able to recover heart rate across most levels of noise (white gaussian noise filter >= 10 dB), even at relatively low sampling rates (>=20 Hz). RapidHRV's recovery of heart rate variability was excellent at relatively low levels of signal to noise ratio (>=20 dB), though there was degradation of performance as sampling rate decreased. Additional simulations of cardiac responses to an anxiety induction demonstrated RapidHRV estimations fully captured effects at moderate levels of noise (>=30 dB) even at relatively low sampling rates (*i.e.*, 20 Hz). RapidHRV was able to partially capture effects (~50% reduction in effect size) at very high levels of noise (>=10 dB when Hz >50). Simulations revealed RapidHRV cleaning was particularly beneficial at lower sampling rates and higher levels of noise, but was not necessary (or could be relaxed) when signal and sampling rates were high. Moreover, these simulations were able to clarify the validity of RapidHRV's default window (10 s) for estimation of heart rate variability across a longer time period (*i.e.*, 5 min).

## Electrocardiography

In our electrocardiography analyses, we were able to recapitulate previously reported effects of age on heart rate and heart rate variability. In line with previous analyses (*Iyengar et al., 1996*), RapidHRV-estimates suggested older participants had lower heart rate variability than younger participants during movie-watching. Importantly, cleaning was vital to this detection.

## Finger PPG

Using RapidHRV estimates, we noted effects of anxiety on heart rate and heart rate variability in a database of participants watching horror and documentary videos while undergoing finger infrared PPG recordings. Notably, the estimated effect size was analogous to that noted in threat-of-shock studies (*Gold, Morey & McCarthy, 2015*). Moreover, when contrasting subject-specific estimates, we found good-to-excellent agreement between RapidHRV and a previous analysis using a commercially available software. Effect sizes between conditions and convergence of estimates between softwares was significantly improved following RapidHRV cleaning.

## Wrist PPG

In our wrist PPG validation, we noted RapidHRV was not able to produce estimates for the majority of conditions due to limitations of the default arguments. Despite applying scaling (0–100) at the level of a sliding window, PPG signals showed variable peaks amplitudes. Following alterations [rhv.analyze(preprocessed, outlier_detection_settings = "conservative", amplitude_threshold = 30, window_overlap = 9)], RapidHRV-estimates during low motion activities (*e.g.*, reading, eating lunch) demonstrated good-to-excellent agreement with a manually-verified analysis of ECG data. During high wrist-movement activities (*e.g.*, table soccer), estimates were generally poor-to-moderate. We do note however that in one of the high motion conditions, cycling, RapidHRV-estimation of heart rate was good (though heart rate variability estimation was poor). This may reflect our

**Table 7 Summary of recommendations for RapidHRV argument inputs across recording modalities.**

| Modality | Relative noise | Min. Amplitude threshold | Outlier rejection | Other |
|---|---|---|---|---|
| ECG | Cleaner | – | 'liberal' | ecg_prt_clustering = True |
| Finger IR PPG (still) | Moderate | 50 (default) | 'moderate' (default) | – |
| Wrist PPG | Noisier | 30 | 'conservative' | window_overlap = 9* |

Note:
* Overlapping windows guarantees non-independence between data points. This should be taken into consideration during statistical testing.

proxy for motion-related artifacts (jerk magnitude), which may not always be as good an estimator of artifacts during activities which involve changes in acceleration but relatively low movements in the wrists (*i.e.*, cycling, as hands are gripping the handle bars). Future work should seek to correlate RapidHRV quality assurance metrics with a wider range of motion estimation methods. Overall, we found adjustments to parameter arguments beneficial to PPG data, but we also note there are contextual factors and limitations (*i.e.*, motion-related artifacts) influencing the feasibility of accurate estimation.

## Overall user recommendations

For ECG data, users may find traditional, amplitude-based analyses will not work for subjects who demonstrate atypical signal morphologies (*e.g.*, particularly prominent P and T waves). RapidHRV includes the use of k-means clustering to help discern these components of the ECG signal, though this is not enabled by default (ecg_prt_clustering = False). Additionally, the cleanliness of signal, namely the stability of peak prominences, means the data may be low in artifacts, and that minor deviations could be detected as outliers. As such, when dealing with already-clean data, users may find that outlier rejection can be omitted or relaxed (*e.g.*, outlier method = 'liberal').

Results from the finger IR PPG data used in the present study did not suggest the need for alterations to default RapidHRV arguments, but did suggest that automated cleaning should be used.

Lastly, PPG data collected from naturalistic settings is typically low in signal to noise ratio, which can constrain peak detection. Consequently, lowering the minimum amplitude threshold for peak detection and decreasing window movement may help improve extraction. Furthermore, given the large amount of motion-related artifacts and the results from our simulation analyses, we recommend: (a) the use of relatively conservative cleaning (*e.g.*, outlier method = 'conservative'), and (b) inspection of motion across conditions as an indicator of estimation accuracy (Table 7).

## CONCLUSION

In the present paper, we have outlined RapidHRV: an open-source Python pipeline for the estimation of heart rate and heart rate variability. Across simulated datasets, RapidHRV showed good-to-excellent recovery of heart rate and heart rate variability at relatively high levels of noise. Estimates in electrocardiography and finger IR PPG demonstrated

RapidHRV was able to recapitulate known effects of age and anxiety, and showed excellent agreement with visually-inspected analyses and commercial software. Lastly, performance in wrist photoplethysmography data was good-to-excellent when participants were engaged in low motion activities, but we noted poor-to-moderate estimations when motion was high. Given the increased interest in the use of wearable measures of heart rate metrics and how they relate to other domains such as mental health, we hope that this toolbox will be of wide use to the community, and that the simulation and benchmarking tests provided may help inform the design and analysis of such studies.

## ACKNOWLEDGEMENTS

Thank you to Kaarina Aho for statistical consultation; Nicolas Legrand for allowing us to use *Systole*'s code for deriving high frequency power; and Russell Kirk for helping derive motion estimates from smartwatch data. Appreciation goes to other open-source analysis software, namely *Systole* (*Legrand & Allen, 2020*) and *HeartPy* (*van Gent et al., 2019*), which helped inspire the development of RapidHRV.

### Funding

This work was supported by the Leverhulme Trust as part of the Doctoral Training Program for the Ecological Study of the Brain (DS-2017-026, to Peter A. Kirk), MRC and NIHR CARP award (MR/V037676/1, to Sarah N. Garfinkel), and MRC Senior Non Clinical Fellowship award (MR/R020817/1, to Oliver J. Robinson). The funders had no role in study design, data collection and analysis, decision to publish, or preparation of the manuscript.

### Grant Disclosures

The following grant information was disclosed by the authors:
Leverhulme Trust as part of the Doctoral Training Program for the Ecological Study of the Brain: DS-2017-026.
MRC and NIHR CARP award: MR/V037676/1.
MRC Senior Non Clinical Fellowship award: MR/R020817/1.

### Competing Interests

Oliver J. Robinson's MRC senior fellowship is partially in collaboration with Cambridge Cognition (who plan to provide in-kind contribution) and he is running an investigator-initiated trial with medication donated by Lundbeck (escitalopram and placebo, no financial contribution). He also holds an MRC-Proximity to discovery award with Roche (who provide in-kind contributions and have sponsored travel for ACP) regarding work on heart rate variability and anxiety. He has also completed consultancy work on affective bias modification for Peak and online CBT for IESO digital health. Oliver J. Robinson sits on the committee of the British Association of Psychopharmacology.

## Author Contributions

- Peter A. Kirk conceived and designed the experiments, performed the experiments, analyzed the data, prepared figures and/or tables, authored or reviewed drafts of the paper, and approved the final draft.
- Alexander Davidson Bryan analyzed the data, prepared figures and/or tables, authored or reviewed drafts of the paper, revised toolbox code, and approved the final draft.
- Sarah N. Garfinkel conceived and designed the experiments, authored or reviewed drafts of the paper, and approved the final draft.
- Oliver J. Robinson conceived and designed the experiments, authored or reviewed drafts of the paper, and approved the final draft.

## Human Ethics

The following information was supplied relating to ethical approvals (*i.e.*, approving body and any reference numbers):

N/a: reanalysis of publicly available data.

## Data Availability

RapidHRV can be downloaded from PyPi (https://pypi.org/project/rapidhrv/) or GitHub (https://github.com/peterakirk/rapidhrv).

Information on the data and code are available at OSF: https://osf.io/7zvn9/.

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
