# Peer review of "RapidHRV: an open-source toolbox for extracting heart rate and heart rate variability"

_PeerJ, doi:10.7717/peerj.13147_

## Round 0.1 · original submission · Major Revisions

Thank you for your submission. The reviewers have provided helpful and detailed comments. Please address these carefully.

Reviewer 1 ·

Basic reporting

- The introduction should go directly to the point (HRV analysis and physiological signal processing, without reviewing the HRV literature in itself).
- The authors should mention the neuroKit2 toolbox and probably cite the following paper, as it is directly related to this area:
Pham, T., Lau, Z. J., Chen, S. H. A., & Makowski, D. (2021). Heart Rate Variability in Psychology: A Review of HRV Indices and an Analysis Tutorial. Sensors, 21(12), 3998. https://doi.org/10.3390/s21123998

- Describe how does the k-mean clustering could help with P and T wave contamination.
- What do the parameters “liberal”, “moderate” and “conservative” refer to? What are the parameters affected?

Experimental design

This paper does not report experimental results but uses open datasets to test the performances of the new methods that were introduced. I think this is nicely done, but I have concerns regarding the way it is tested.

- It is not clear to me that the different tests detailed in the paper demonstrate the accuracy of the package. The authors often state that they found “excellent” or “good-to-excellent” agreements, but these statements are not grounded on model comparison. Measuring the error of the estimate (eg. RMSE) and comparing this with other toolboxes, or ground truth might be more appropriate.
- One of the originalities of this toolbox is a new peaks detection algorithm based on clustering. The performance of this algorithm should be directly compared to the state-of-the-art algorithms for the same task (see e.g. https://github.com/berndporr/py-ecg-detectors for ECG). As other tools are already provided in the Python ecosystem, it would be relevant here to compare the PPG performances with NeuroKit2 and/or Systole, as it is not clear for now that this pipeline performs better, we just see that it is not performing badly.

Validity of the findings

This paper introduces a new open-sourced Python pipeline to perform heart rate variability analyses in an easy and reproducible way. Heart rate variability is widely used in psychology, physiology, and neuroscience in general and, as Python itself is gaining popularity in these fields, this contribution will certainly find a large range of potential users. There are some alternatives already available in the same area. This toolbox introduces original components but it might be too limited to be considered as a package in itself. The paper also details several analyses that are interesting from a method point of view, but the pipeline proposed is debatable and not really tested against other algorithms.
I think the following points that concern the novelty of this work, and the way the package is tested should be addressed before this paper can be considered for publication.

- This paper introduces rapidHRV as a heart rate variability toolbox, however, the range of metrics this package can compute is extremely limited. I understand that the HRV literature already incorporates hundreds of mathematical derivations of variability, and including all of them in every package might not be necessary. But some gold standards are critically missing (e.g. frequency domain and non-linear domain). The authors should point to other packages that can be used to extract such features, as only extracting the RMSSD from a given recording might appear extremely limiting. It also questions the name of the package itself, as rapidHRV would suggest something oriented toward heart rate variability in general, this module could be renamed rapidRMSSD? I do not think that the package should be renamed, I think its scope should be enlarged.

Additional comments

Some of my comments concern the codebase in itself (as it can be found in https://github.com/peterakirk/RapidHRV):
- I am a bit concern that this package does not include any test suite (as far as I can see). I think this is the main problem, for now. The code should be tested, ideally, this should be performed automatically using Github action or something similar (TravisCI …). The code coverage should be estimated from this test suite and should be maximized.
- Providing example data set that can be automatically downloaded using some function would make it easier to test the code and run tutorials (right now, running the code snippet from the Readme returns an error as the variable `mydata` is not defined).
- It would be convenient to have tutorials provided as notebooks, making it run on Google Colab can also increase readability and help users test the package before installing it.
- The code should follow clear formatting rules (flake8, black…). Providing type hints and static checks can be a plus.
- The install requires the exact version of packages, which is a bit brutal for the user and probably not necessary, I would recommend version >= something instead wherever possible.
- The API could be simplified (e.g. rapidhrv.preprocess.preprocess could be shortened into rapidhrv.preprocess for import)

·

Basic reporting

This manuscript by Kirk and colleagues presents and describes RapidHRV as an open-source Python toolbox for extracting heart rate and time-domain heart rate variability measures. It does so by introducing the background and the motivation of the package. The authors also present different testing scenarios, using simulated and real datasets against different signal-to-noise ratios and different sampling rates, and with signals of different modalities (ECG, PPG, and pulse oximetry).
I think this toolbox is within the scope of PeerJ and will be of interest to the readers even though some issues would need to be addressed before publication. I recommend "major revision" to have the possibility to read the modified version. As part of this review, I have looked at the source code of the package in the GitHub repository, tried to install and run the example scripts. You'll find below my suggestions and questions. Note that authors are free to not take them into an account provided they explain why.

Experimental design

no comment

Validity of the findings

no comment

Additional comments

General comments:

• I was able to install the package and run the example found in the GitHub repository. I commend the authors for putting together the documentation for the functions and the arguments in the docstrings which greatly help with the readability of the code. However, I have a few comments about the styling of the code and the documentation which can be found below.

• While the authors did briefly mention other toolboxes for HRV in the introduction (line 51), they have missed out on some toolboxes that are also open-source and have been developed to rigorously work with noisy data such as:
o pyHRV: https://github.com/PGomes92/pyhrv
o NeuroKit2: https://github.com/neuropsychology/NeuroKit

I recommend the authors further elaborate on the advantages of RapidHRV compared to these existing packages (heartPy, pyHRV, NeuroKit2, hrv) and how the proposed pipeline in RapidHRV would work more rigorously with data from wearable devices. I understand that doing a benchmarking study against these packages would be a lot of work and might be out of scope for this manuscript. However, it would be good to acknowledge the existing packages, how are they different/similar to RapidHRV, and perhaps discuss a few features that are uniquely RapidHRV (e.g., clustering of QRS waves).

Major issues:
• K-mean clustering to discern different QRS features seems to be a unique feature of RapidHRV. I would recommend the authors to illustrate with an example when and how it can be useful. If there is literature to support this step, do include it too. And if adding an example would disrupt the flow of the current manuscript, the authors can do so in the GitHub repository and refer to the example accordingly.

• As the toolbox specifically focuses on the time-domain features, I believe that having only 3 measures (BPM, RMSSD, and SDNN) would significantly limit the usage of the package. Perhaps the authors can discuss a future plan to include more of them? If you are looking for a review of different HRV indices, you can consider our recent work: https://www.mdpi.com/1424-8220/21/12/3998, which includes an exhaustive list of time-domain measures. Nevertheless, please feel free to refer to any other suitable works.

• “Time from the first peak to the last peak does not recede 50% of the fixed window width. This is to ensure the user that the actual length of time for extracting BPM/RMSSD is not less than half of that which is specified in the window width argument.” (line 152). The relevance and usefulness of this statistical constraint are not as clear as the others. Could the authors elaborate a little more?

• Comments about the software:
o Unit-test to maintain package: currently, the functions are not being tested (using pytest or Travis). Having the functions tested and benchmarked helps to maintain the stability of the package (e.g., functions can break due to updates in other dependencies). Thus, the authors are recommended to add basic tests, at least for the three main functions, to ensure the quality of the code.
o Plot legends in visualization: authors should add matplotlib legends to specify to users what do the “X”, the black, grey, and purple dashed lines represent in the plot. We were able to read Figure 3, thanks to its legend. However, for your future users, plot legends should be added so that the information in the plot can be conveyed clearly.

Minor issues:

• While the normalization steps in the manuscript are described as
“Normalization is performed at the level of a sliding window during the analysis stage (‘extract_heart’ module), wherein the mean signal is subtracted from each data point in a given window, then multiplied by 100. Finally, this is divided by the range of values (maximum - minimum data point in the window). “,
the source code in the GitHub repository for normalization is:
“(data - np.min(data)) * 100 / (np.max(data) - np.min(data))”.
There seems to be a discrepancy here. It seems like the authors would like to perform the min-max normalization (as described in the source code) and had made a typo in the manuscript.

• Since normalization is a step performed in the analysis stage, maybe the authors could consider describing them in the later paragraph, instead of having them together with the pre-processing steps (line 79).

• The caption of Table 2 should specify what is the value inside and outside of the brackets.

• Code styling and readability:
o For code readability, authors should consider following the PEP guidelines (https://www.python.org/dev/peps/pep-0008/). For instance, underscores can be used in the argument names to separate the words (e.g., `input_data` instead of `inputdata`). Nevertheless, this is a minor comment about styling and needs not to be addressed by the authors for the manuscript to be considered for publication.
o Authors should also follow the standard recommendations for readable docstring syntax. The common ones are google docstrings (https://google.github.io/styleguide/pyguide.html#38-comments-and-docstrings) or numpydoc (https://numpydoc.readthedocs.io/en/latest/format.html). Again, this comment about styling is to help the package to have better readability and thus should not be critical for the decision for publication.

---

## Round 0.2 · Minor Revisions

I heard back from one of the reviewers. Overall, they were happy with the changes but requested that you check "the citations one last time before publishing (e.g. the Pandas citation is not the one recommended on the website)." Please resubmit after having done that.

Reviewer 1 ·

Basic reporting

No comment

Experimental design

No comment

Validity of the findings

No comment

Additional comments

The authors have made all the changes that were requested in the first review and I do not have additional comments to make on the current version of the paper.

I would only recommend checking the citations one last time before publishing (e.g. the Pandas citation is not the one recommended on the website).

Congratulation to the authors for their work and their efforts in the review process.

---

## Round 0.3 · accepted · Accept

Thank you for checking that reference.